# Genomic Dissection through Whole-Genome Resequencing of Five Local Pig Breeds from Shanghai, China

**DOI:** 10.3390/ani13233727

**Published:** 2023-12-01

**Authors:** Jun Gao, Lingwei Sun, Hongmei Pan, Shushan Zhang, Jiehuan Xu, Mengqian He, Keqing Zhang, Jinyong Zhou, Defu Zhang, Caifeng Wu, Jianjun Dai

**Affiliations:** 1Institute of Animal Husbandry and Veterinary Science, Shanghai Academy of Agricultural Sciences, Shanghai 201106, China; gaojun@saas.sh.cn (J.G.); sunlingwei1987@126.com (L.S.); smalltreexj@126.com (S.Z.); jiehuanxu810@163.com (J.X.); he1037247863@163.com (M.H.); zhangdefuzdf@163.com (D.Z.); 2Shanghai Municipal Key Laboratory of Agri-Genetics and Breeding, Shanghai 201106, China; 18141634280zk@gmail.com (K.Z.); jinyzhou@126.com (J.Z.); 3Key Laboratory of Livestock and Poultry Resources Evaluation and Utilization, Ministry of Agriculture and Rural Affairs, Shanghai 201106, China; 4Chongqing Academy of Animal Sciences, Chongqing 402460, China; panhm_2118@163.com

**Keywords:** indigenous Shanghai pig breed, commercial pig breed, whole-genome resequencing, selective sweep analysis, genomic characteristics

## Abstract

**Simple Summary:**

Whole-genome resequencing was performed on five Shanghai local pig breeds, aiming to analyze their population’s genetic structure and genomic characteristics. Through selective sweep analysis between the indigenous Shanghai pig breed and commercial pig breed populations, some of the selection genes and their polymorphisms in these two populations were identified and explored.

**Abstract:**

China has rich genetic resources of local pig breeds. In this study, whole-genome resequencing was performed on five Shanghai local pig breeds, aiming to analyze their population genetic structure and unique genomic characteristics. Tens of millions of single nucleotide variants were obtained through the resequencing of a total of 150 individual pigs from five local pig breeds (Meishan, Fengjing, Shawutou, Pudong White, and Shanghai White) after mapping them with the pig reference genome of *Sus scrofa* 11.1. The results of admixture structure analysis also clearly demonstrated the genetic differences between the Shanghai local pig breeds and the three commercial pig breeds (Duroc, Landrace, and Yorkshire). The genetic infiltration of Landrace and Yorkshire pig breeds in the SHW breed was detected, which is consistent with the early history of crossbreeding in this breed. Selective sweep analysis between four indigenous Shanghai pig breed populations and three commercial pig breed populations identified 270 and 224 genes with selective signatures in the commercial and indigenous Shanghai pig populations, respectively. Six genes (*TGS1*, *PLAG1*, *CHCHD7*, *LCORL*, *TMEM68*, and *TMEM8B*) were found to be associated with animal growth in the commercial pig population through gene enrichment and protein–protein interaction analysis. In contrast, the *MSRB3* gene in the indigenous Shanghai pig population was significantly under selection, which correlated with the long pendulous ear phenotype of the indigenous Shanghai pig population. In conclusion, this study is the first genomic profiling of five representative local pig breeds in Shanghai, which provides molecular genetic data and foundations for better conservation and utilization of local pig breed resources in Shanghai, China.

## 1. Introduction

Phenotype variation and genomic differences between cosmopolitan and indigenous pig breeds are due to natural and artificial directional selection, which shapes the reservoir of animal genetic diversity. Shanghai local pig breeds have a long history of domestication. However, with the urbanization process, only four indigenous breeds have been preserved, and only in a few designated breeding farms. These include the national middle-sized Meishan pig (MMS) breeding farm in Jiading district, the Fengjing pig (FJ) breeding farm in Jinshan district, the Pudong White pig (PD) breeding farm in the new Pudong district, and the Shawutou pig (SW) breeding farm on Chongming Island. In addition to these four indigenous Shanghai pig breeds, there is also a Shanghai White pig (SHW) breed, which was bred by crossing the Shanghai local breed with the Landrace, Yorkshire, and Jiangsu White pig in early years and passed breed validation in 1978, and is currently kept in the breeding farm of the Zhuanghang Experimental Station of Shanghai Academy of Agricultural Sciences (Figure 1).

The genetic diversity and unique genomic characteristics of Shanghai local pig breeds are valuable biological heritage, which can be used to discover superior traits of pig breeds as well as to improve genetic breeding. Although indigenous pig breeds are not as good as cosmopolitan breeds and strains used in intensive production systems in terms of growth rate and feed conversion rate, these autochthonous genetic resources have unique advantages in terms of certain traits; for example, Meishan pigs have been widely used for the improvement of many commercial breeds [1], and its reference genome was also assembled and released in 2021 [2]. Most Asian breeds exhibit higher prolificacy, more fat accumulation, and slower growth than European pigs [2,3,4,5]. On the other hand, local pig breeds have a good niche market due to their unique taste and flavor. However, to the best of our knowledge, the genomic characteristics of these representative local pig breeds in Shanghai have not been fully studied.

The selected region is often a chromosomal region with low genetic diversity within the group and a high genetic differentiation rate between groups. A series of statistical approaches called “classical selective sweeps” have been proposed for the detection of “signatures of selection” [6,7], such as genetic differentiation value (F_ST_), polymorphism level statistic (θ_π_), extended haplotype homozygosity (EHH) [8], integrated haplotype homozygosity score (iHS) [9], and composite likelihood ratio (CLR) [10]. The method most frequently used to assess population structure is the calculation of F_ST_, which was first introduced by Sewall Wright [11] and developed by Weir and Cockerham [12]. The polymorphism level (θ_π_) statistic was measured for each individual using nucleotide diversity π [13,14], usually utilized to identify regions of selection between the different populations. The investigation of genomic regions with F_ST_ and θ_π_ crossover methods helps detect selection features and has been successfully applied to identify genomically selected regions in domestic pigs [15,16,17].

In this study, we performed whole-genome resequencing on the abovementioned five representative local pig breeds. Through genome-wide genetic variation analysis, aimed to explore the genomic characteristics of Shanghai local pig breeds, the genes with selective signatures and the association of related phenotypes are explored through comparative analysis with the cosmopolitan commercial pig population.

## 2. Materials and Methods

### 2.1. Sample Collection and Sequencing

The blood samples collected from designated breeding farms of five local breeds in Shanghai (29–31 individual pigs were selected from each breed, for a total of 150 individual pigs, covering all recorded pedigrees of these five breeds, respectively, with a male to female ratio of 1:2 and exclusion of individual pigs from the same litter) and whole-genome sequencing (WGS) data from 31 Duroc, 21 Landrace, and 24 Yorkshire pigs downloaded from the NCBI database (https://www.ncbi.nlm.nih.gov/ (accessed on 16 December 2022)) were placed together in Appendix A. Genomic DNA was extracted from blood samples using a standard phenol–chloroform method [18] and stored at 4 °C to avoid freeze-thawing, and DNA concentration (ng/mL) and quality were determined using Nanodrop and agarose electrophoresis. DNA was fragmented and processed according to GenoBaits^®^ DNA Library Prep Kit for ILM (MolBreeding Biotech Company, Shijiazhuang, China), constructed from resequencing libraries from QC-qualified DNA, proceeding as follows: end repair, A-tailing, ligation to paired aptamers, and PCR amplification with a 300–350 bp insert. After the library construction was completed, Qubit 2.0 was used for preliminary quantification, and qPCR was used to accurately quantify the effective concentration of the library to ensure the quality of the library. Sequencing was performed using the MGI-2000 sequencing platform (BGI, Shenzhen, China), and the sequencing mode was PE150 mode.

### 2.2. Variant Calling and Annotation

The raw reads were filtered using the software fastp [19] (version 0.20.0, parameters: -n 10 -q 20 -u 40) to obtain clean reads. Clean sequencing reads were mapped to the reference genome *Sus scrofa* 11.1 using Burrows-Wheeler Aligner (BWA) v0.7.17. [20] with default settings. The Genome Analysis Toolkit (GATK) v4.0 [21] was used for single nucleotide variant (SNV) calling. The HaplotypeCaller module performed variant detection with the following parameters: GATK Best Practices. Filtering was performed using the VariantFiltration module with filter parameters: --filter-expression “QD < 2.0||QUAL < 30.0||MQ < 40.0||FS > 60.0||SOR > 3.0||MQRankSum < 12.5||ReadPosRankSum <−8.0”. Finally, ANNOVAR was used to conduct gene-based or region-based annotation processing for the filtered variants [22], and the corresponding gene annotation file was downloaded from the Ensembl database (https://asia.ensembl.org/index.html (accessed on 23 January 2023)).

### 2.3. Genetic Analysis of the Population

All SNPs located on the autosomes (Chr1-18) of the eight breeds were controlled by a multiple filtering process using Plink v1.9, with parameters such as missing rate (--mind 0.1), minimum allele frequency (--maf 0.05), Hardy–Weinberg equilibrium (--hwe 0.000001), and LD pruning (--indep-pairwise 100 50 0.2) being filtered, and the final retained SNPs were used for the analysis of the population structure. Principal component analysis (PCA) was performed via Plink v1.9 software, in which the principal components were calculated with “--pca” option. The PCA plot demonstrated the first two principal components using the ggplot2 package [23] in the R program. The population ancestry of these breeds was inferred using ADMIXTURE software v1.3.0 [24]. The ancestral clusters K were tested from 2 to 10 with a five-fold cross-validation to explore the optimum ancestral groups. The identity-by-state (IBS) matrix was calculated using Plink v1.9 software [25] and a neighbor-joining (NJ) phylogenetic tree was constructed based on the IBS distance matrix using PHYLIP software v3.698 [26]. Finally, the NJ tree was visualized via iTOL v6 (https://itol.embl.de/ (accessed on 21 June 2023)) [27].

### 2.4. Genome-Wide Selective Sweeps Detection

Based on the results of the genetic structure analysis of the eight breeds mentioned above, we decided to identify some potential selection signals in the polymorphism level of the genomic regions of the Shanghai native pig herd (120 individual pigs from the four indigenous Shanghai breeds, “MMS”, “FJ”, “SW”, and “PD”) and the commercial herd (76 individual pigs from the Duroc, Long White, and Yorkshire breeds). A crossover approach was used to calculate F_ST_ and θ_π_ ratios on all autosomes and X chromosomes using a 40 kb sliding window approach with a step size of 20 kb by aligning genomic windows to match the top 5% F_ST_ and combining θ_π_ ratios (indigenous Shanghai population/commercial pig population), taking the log2-transformed top 5% percentile at each end of the overlapping windows and using them as candidate selective regions. The candidate genes were subsequently extracted using bcftools v1.10.2 [28]. The gene annotation files and the polymorphism levels upstream and downstream of the candidate genome were investigated.

### 2.5. Enrichment Analysis and PPI Network Construction

The candidate genes in these selection regions were identified through assembly (http://ftp.esembl.org/pub/curent_gtf/sus_scrofa/Sus_scrofa.Sscrofa11.1.106.gtf.gz (accessed on 8 February 2023)). The biological process of Gene Ontology (GO) terms and the Kyoto Encyclopedia of Genes and Genomes (KEGG) pathway enrichment analyses were performed using the WEB-based Gene Set Analysis Toolkit (www.webgestalt.org (accessed on 15 June 2023)) [29]. Next, to further study the interaction associations of these selection genes, the STRING (Search Tool for the Retrieval of Interacting Genes/Proteins) database (https://string-db.org/ (accessed on 25 June 2023)) [30], which contains known and predicted PPI information by consolidating known and predicted protein–protein association data for a large number of organisms, was applied for PPI analysis. The selection genes were mapped, and the interactions with default confidence parameters of 0.4 were used. The protein–protein interaction (PPI) network was constructed, and Markov Cluster Algorithm (MCL) clustering was used for subnetwork construction with the inflation parameter of default 3.

## 3. Results

### 3.1. Genomic Variant Identification in Shanghai Local Breeds

Whole-genome resequencing of 150 individual pigs from five pig breeds (MMS, FJ, SW, PD, and SHW) generated a total size of 3.42 Tb clean data, which was deposited in the publicly accessible website: http://www.cncb.ac.cn (accessed on 17 October 2023) (China National Center for Bioinformation, CNCB) [31,32] with BioProject PRJCA020140 with GSA number CRA012827. After quality control, genomes of the above 226 pigs were lined up against the *Sus scrofa* 11.1 reference genome, resulting in an average depth of 12–15.25 folds (Table 1). A total of 40,874,009 SNPs and 14,859,192 InDels were detected, respectively. Based on the location of the variant locus on the reference genome, the intergenic variant accounted for 45.7%, intronic variants accounted for 42.2%, and exonic variants accounted for 0.7% of the of the total variants.

### 3.2. Population Genetic Structure

After multiple screening of all SNPs on the autosomes (Chr1-18) of the eight pig breeds for missing rate, minimum allele frequency, and linkage disequilibrium (LD) pruning using Plink v1.9, 223 individuals and 569,301 SNPs were retained for population structure analysis. The PCA results showed that three foreign breeds (Duroc, Landrace, and Yorkshire) were clustered together. Three indigenous Shanghai pig breeds (MMS, FJ, and SW) were also clustered together, while the indigenous Pudong White Pig breed (PD) was clustered separately (Figure 2A). The population structure, based on these 569 K SNPs, was inferred by Admixture (K from 2 to 10) that it was appropriate to classify all individuals into eight populations because of the low cross-validation error obtained (K = 8, Figure 2B). It was evident that among all the individual pigs, only the Landrace and Yorkshire individuals showed obvious genetic infiltration, and several of the individuals in the SHW population also indicated a genetic infiltration from the Landrace; meanwhile, there was no obvious interbreed gene infiltration among the individuals of the other four indigenous Shanghai pig breeds (Figure 2C).

The IBS matrix among the individuals also showed results consistent with the above admixture’s inferred results. The three foreign breeds (Duroc, Landrace, and Yorkshire) were relatively close to the SHW breed with relatedly low IBS distance (Figure 3A). On the contrary, the IBS distances between the four indigenous Shanghai pig breeds were small, while that between them and the three commercial pig breeds were large. The neighbor-joining (NJ) phylogenetic tree demonstrated that there were no individual clustering errors between different breeds except for one Yorkshire individual clustered with the Landrace group (Figure 3B).

### 3.3. Selection Signature Detection of Indigenous Shanghai Pig Population

The individual SHW pigs were removed in the selection sweep analysis since this breed was bred by crossbreeding Shanghai native pigs with foreign breeds in the early years, and the results of the genetic structure analysis confirmed that some of the individuals in the population had obvious genetic infiltrations of Landrace and Yorkshire; thus, we removed this breed in the analysis of the selection signals.

A total of 119,545 windows were obtained on the pig genome (excluding the Y chromosome) by using 40 K length as the genomic sliding window and 20 K as the step. Genome-wide selection signals were investigated by aligning all windows and using two complementary approaches, genetic differentiation (F_ST_) and polymorphism level (θ*_π_*). To minimize false-positive candidate regions, regions that reached the top 5% threshold for both methods were selected as candidate regions. The results of the study showed that 542 genomic regions under selection were identified as belonging to the Shanghai indigenous population (threshold, 5%; F_ST_, 0.606245; Log_2_θ*_π_* ratio, < −0.677740, blue scatters in Figure 4), whereas 888 genomic regions under selection pressure were identified as belonging to the commercial pig population (the same F_ST_ threshold and log_2_θ*_π_* ratio > 2.243567, red scatter in Figure 4). Detailed analysis data were placed in Appendix A.

### 3.4. Functional Analysis of Genes in Selected Regions

The annotated genes within the selected regions screened above were extracted, and a total of 270 selected genes were identified in the commercial pig breeds, while 224 selection genes were identified in the Shanghai local pig breeds. Information on these genes was placed in Appendix A. The 270 selection genes in the commercial pig population represent a significant decrease in polymorphisms of these genes in the commercial pig population compared to the indigenous Shanghai pig population. Similarly, the 224 selection genes in the indigenous Shanghai pig population represent a significant decrease in polymorphism of these genes in the indigenous Shanghai pig population.

Enrichment analysis of the gene sets (GO/KEGG) revealed that pathways such as ovarian steroidogenesis and steroid hormone biosyntheses were enriched (Figure 5 and Appendix A), and that both pathways contained *CYP19A1* and *CYP19A3*. By investigating the polymorphism of *CYP19A1*, a total of 456 SNPs were identified within the gene as well as upstream and downstream among the individual pigs, of which two mutations were located in the coding region (Chr1_120657973, T/C, synonymous variant and Chr1_120695729, G/A, missense variant), whereas these two loci were pure reference genome genotypes in 76 commercial pig breeds (0/0), and higher polymorphisms existed in four indigenous pig breeds in Shanghai; for example, high polymorphisms in these two loci were found in the SW breed, with 25 out of 31 individual pigs and 19 out of 31 individual pigs, respectively.

### 3.5. Protein–Protein Interaction (PPI) Subnetwork

After analyzing the PPI of 270 selected genes identified in commercial pigs and 224 genes detected in the indigenous Shanghai pig population, a total of 68 PPI subnetworks were obtained for the 270 genes in commercial pigs. One PPI subnetwork of six genes (Figure 6A) attracted our attention because the associated literature showed that the genes in this network have been demonstrated in multiple studies of selection signaling in livestock animals and are strongly correlated with the body size of livestock animals (Figure 6C). For example, three proteins, CHCHD7, PLAG1, and LCORL, were mentioned in the 2018 study of large- and small-bodied pigs in PMID: 30231878. These three genes were also mentioned in the 2013 GWAS study of birth weight in Nellore cattle PMID: 23758625. Two proteins, PLAG and LCORL, were also mentioned in the 2012 genome-wide study of selection signaling in pigs (PMID: 23151514).

On the other hand, from the PPI analysis of 224 genes selected from the indigenous Shanghai pig population, a subnetwork of three genes (Figure 6B) is noteworthy because the three proteins are thought to be closely related to the animal’s ear phenotype (Figure 6D). For example, in PMID: 28407177, the mRNA and protein expression levels of these three genes were found to correlate with the ear size of Erhualian pigs and large white pigs (Yorkshire pigs) from the Taihu Lake region of China. Similarly, these three genes have also been reported in several studies of selection signaling in livestock animals, such as pig, cattle, and sheep in the Figure 6D.

Our study further investigated the differences in SNP polymorphisms (a total of 27 SNPs detected) in the *CHCHD7* on chromosome 4 between the two populations, which revealed that all 27 SNP loci were pure reference genome genotypes in foreign pigs such as Duroc, Landrace, and Yorkshire, whereas significant polymorphisms were demonstrated in four indigenous Shanghai pig breeds such as the Meishan pig, of which about two-thirds were heterozygous 0/1 or pure 1/1 genotypes.

Similarly, we investigated polymorphisms in three genes that have been subjected to selection in indigenous Shanghai pigs, particularly the *MSRB3* that has been suggested by several studies to be potentially associated with the trait of long pendulous ears in animals. In contrast, all four indigenous pig breeds in Shanghai have a distinctive long pendulous ear phenotype (Figure 1). Therefore, we investigated the polymorphism of this gene in eight pig breeds.

### 3.6. Polymorphisms of MSRB3 within Indigenous Shanghai Population

The *MSRB3* is located at position 29695824–29863601 on chromosome 5 of the *Sus scrofa* 11.1 reference genome. The nucleic acid polymorphism (π) of this gene showed a significant decrease in the indigenous Shanghai pig breed population compared to the commercial pig population (Figure 7A). A total of 386 SNP sites were detected in *MSRB3* and upstream and downstream in 223 individuals of the eight breeds in this study. A haplotype heatmap of *MSRB3* was constructed by combining these 386 SNPs as columns and the sample individuals as rows (Figure 7B); the colored sections in pink show purely syntenic sites that are consistent with polymorphisms in the reference genome (0/0), whereas sites that present polymorphic loci (0/1 and 1/1) are labeled in blue. From the figure, we can clearly see that almost all four indigenous Shanghai pig breeds have pink pure loci, whereas most individuals in the commercial pig breeds of Yorkshire and Shanghai-bred SHW have a different genotype (blue) for these SNP loci. In contrast, a few individuals in the Yorkshire and SHW breeds of the eight pig breeds had a vertical ear phenotype. In addition, we also observed four neighboring SNP loci (marked by red dashed boxes in Figure 7B), which seemed to characteristically appear in the four indigenous Shanghai pig populations, while these four loci were instead less polymorphic in other commercial pig breeds and SHW breeds. The four neighboring polymorphic sites were located on chromosome 5 at positions 29786809, 29786832, 29786919, and 29786949, all of which were genetic variants of introns. Among them, the 29786832 locus (GA to G, an A-base deletion in an indigenous Shanghai pig population) was the most distinctive feature (detailed polymorphism information had been put in Appendix A).

## 4. Discussion

Local Chinese pigs have rich genetic diversity and are an important part of the genetic resources of pig breeds in the world. In recent years, the conservation of local pig breeds in Shanghai has received increasing attention, and the five pig breeds in this study were included in the Shanghai Livestock and Poultry Resources Conservation Catalog, which contains a total of 11 breeds of livestock and poultry, of which the five pig breeds in this study account for nearly half, indicating the importance and representativeness of the five breeds. As the supporting organization of the Shanghai Livestock and Poultry Genetic Resources Gene Bank, our laboratory has been engaged in the preservation of local breeds for a long time. This study is the first to systematically analyze the molecular genetic structure of local Shanghai pig breeds and their genomic characteristics.

This study revealed the genetic structure of local Shanghai pig breeds, in which four indigenous Shanghai breeds showed significant differences from commercial pig breeds. The SHW population, which was bred through crossbreeding in the 1970s, shows significant gene infiltration from Landrace and Yorkshire genetic backgrounds, which is consistent with the early crossbreeding process. The results of selective sweep analysis of the four indigenous Shanghai breed populations and the cosmopolitan population, respectively, suggested that hundreds of genes under selection, and the linkage between these genes and the phenotypic traits of the animals, are key candidates for research on the molecular genetics of local Shanghai pig breeds. For example, *CYP19A1* and *CYP19A3*, which are closely linked to reproductive traits, are abundantly polymorphic in indigenous Shanghai pig breeds, while they show loss of polymorphism in commercial pig populations. The reported variants in *CYP19A1* can affect in vitro embryo production traits in cattle [33]. The proposed signal for maternal recognition of pregnancy in pigs is estrogen (E2); a loss-of-function study was conducted by editing aromatase (CYP19A1) using CRISPR/Cas9 technology [34]. Despite the loss of conceptus E2 production, CYP19A1 −/− conceptuses were capable of maintaining the corpora lutea. However, gilts gestating CYP19A1 −/− embryos aborted between days 27 and 31 of gestation. On the other hand, the regulatory network formed by the six genes (*TGS1*, *PLAG1*, *CHCHD7*, *LCORL*, *TMEM68*, and *TMEM8B*) identified in this study that are associated with growth or body size of animals [35,36,37] are also worthy of further study because they have rich genetic diversity in the indigenous Shanghai pig population compared to the commercial pig population, and the upstream and downstream regulatory regions in these genes or the SNVs on the expression of these genes is the focus of subsequent studies.

One of the more obvious common features of the appearance phenotype of the four indigenous pig breeds in Shanghai is the possession of a long pendulous large ear phenotype, and the features shown on the genome suggest three related genes including *MSRB3*, *WIF1*, and *LEMD3*, which are located adjacent to each other on the genome. Boyko et al. [38] conducted a genome-wide association study using genomic microarray data from 915 domestic dogs (80 breeds), 83 gray wolves, 10 African dogs, and phenotypic data for 57 traits; finally, it was determined that the 100 kb region upstream of the foot–foot gene was significantly associated with the droopy-ear phenotype of the domestic dogs. Zhang et al. [39] found that the mRNA expression of *MSRB3* was significantly higher in 60-day-old Minzhu pigs than in Large White pigs of the same age, and that individual pigs with higher expression had larger ears. Evidence at the protein level indicated that *MSRB3* expression was higher in 60-day-old two-flower-face pigs than in same-day-old Large White pigs. Chen et al. [40] identified a 38.7 kb copy number variant (CNV) affecting the last two exons of *MSRB3*, which was only present in six native Chinese Long Pendulous and Half Pendulous pigs. The experiment also showed that miR-584-5p inhibited the mRNA translation of *MSRB3*, which led to the enlargement of pig ears.

In this study, the nucleotide polymorphisms of *MSRB3* were significantly decreased in the four indigenous Shanghai pig breed groups in this study compared with the commercial pig breeds. In contrast, some of the SHW pigs showed similar genotypes to some individuals in commercial pigs, especially Yorkshire pigs, many of which have erect ears. And we also visually inspected part of the SHW population, in which there are some individual pigs with an erect ear phenotype. Therefore, there should be a clear link between *MSRB3* and the morphology of pig ears, but the mechanism remains to be further elucidated. On the other hand, although the overall polymorphism of *MSRB3* was significantly reduced in the indigenous Shanghai pig population, some of the loci showed high polymorphism on the contrary; for example, the four adjacent intron loci found in this study, which we analyzed were possibly related to the evolution of pig breeds, are also part of the genomic characteristics of the indigenous Shanghai pig breeds, which can possibly be used as genetic markers for breed identification in the future.

## 5. Conclusions

In this study, whole-genome resequencing was performed on five local Shanghai pig breeds, and information on tens of millions of single nucleotide variants on the genomes of these local Shanghai pig breeds was obtained. Selective sweep analysis revealed that selection genes such as *TGS1*, *PLAG1*, *CHCHD7*, *LCORL*, *TMEM68*, and *TMEM8B* are closely related to animal growth traits and had higher polymorphisms in the indigenous Shanghai pig population However, the *MSRB3* gene, which is linked to the long pendulous ear phenotype, showed a significant characterization of reduction in genetic diversity in the indigenous Shanghai pig breeds. Our study provides molecular genetic data and a foundation for better conservation and utilization of local pig breed resources in Shanghai, China.

## Figures and Tables

**Figure 1 animals-13-03727-f001:**
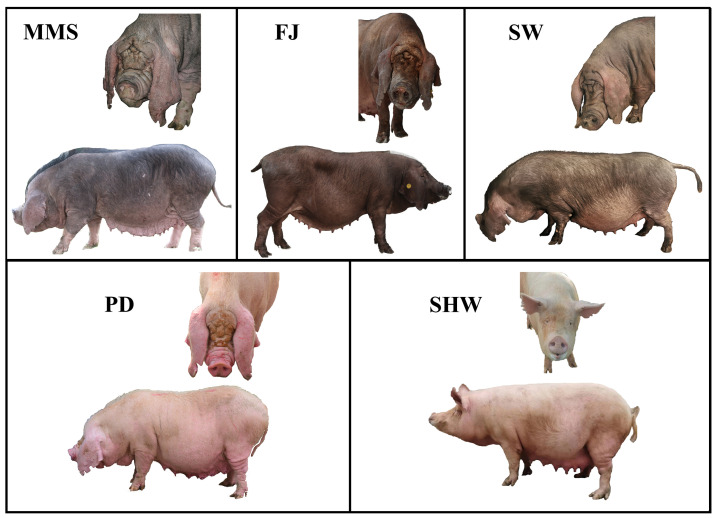
Appearance of five Shanghai local pig breeds.

**Figure 2 animals-13-03727-f002:**
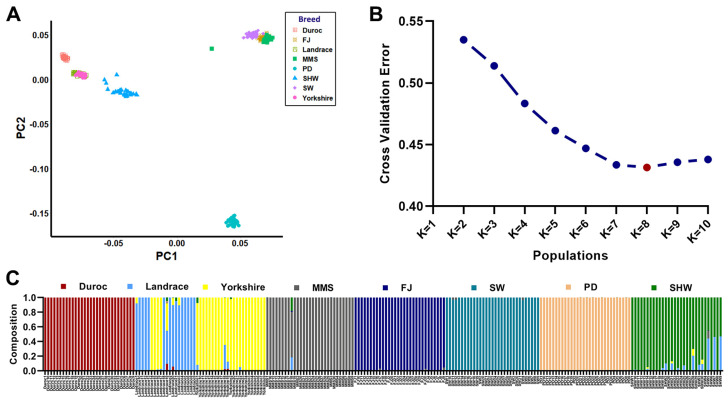
Population structure of the eight pig breeds. (**A**) PCA analysis of the eight pig breeds. (**B**) The lowest cross-validation error rate for all samples divided into eight populations (K = 8, red dot). (**C**) Admixture population structure (K = 8) of the eight pig breeds.

**Figure 3 animals-13-03727-f003:**
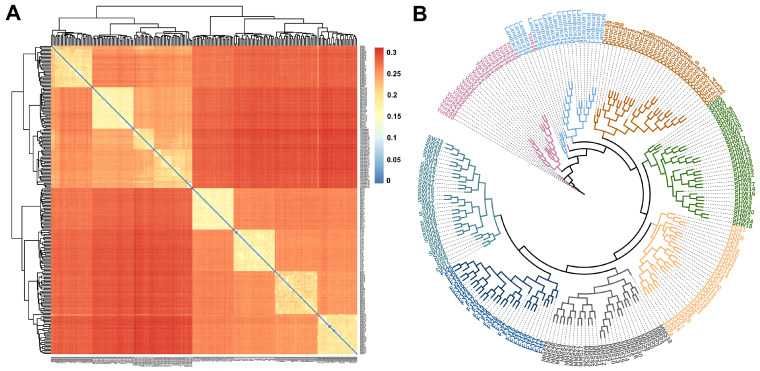
Heatmaps of IBS distances and neighbor-joining (NJ) trees among individuals. (**A**) Heatmaps of IBS distances of the eight pig breed individuals. (**B**) The NJ tree among the eight pig breed individuals.

**Figure 4 animals-13-03727-f004:**
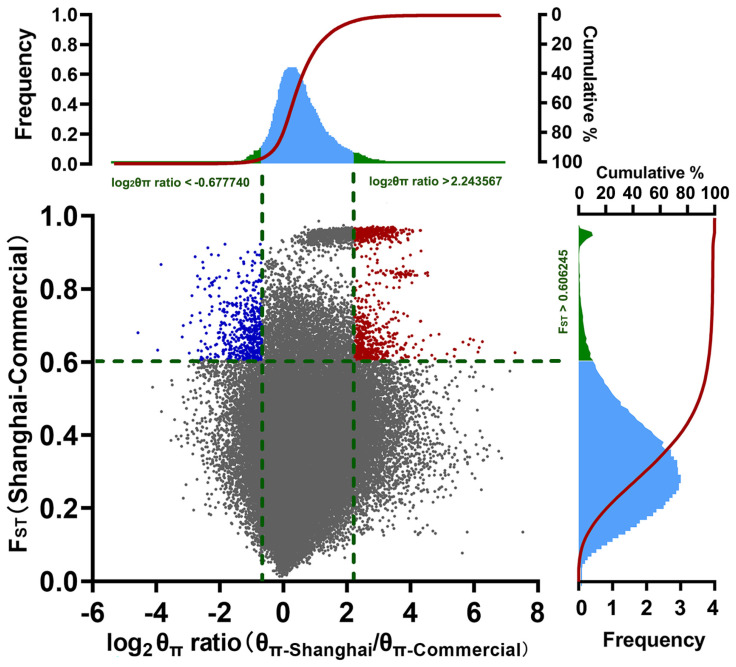
Genome-wide distribution of selection signatures detected using F_ST_ and log_2_θ_π_ ratio. The X-axis represents the log2 normalized value of the θ*_π_* value ratio, and the Y-axis represents the F_ST_ value. The red line displays the threshold level of 5%. The blue scatters indicate the genomic regions under selection in the Shanghai indigenous population, while red scatters revealed genomic selection regions in the commercial pig population.

**Figure 5 animals-13-03727-f005:**
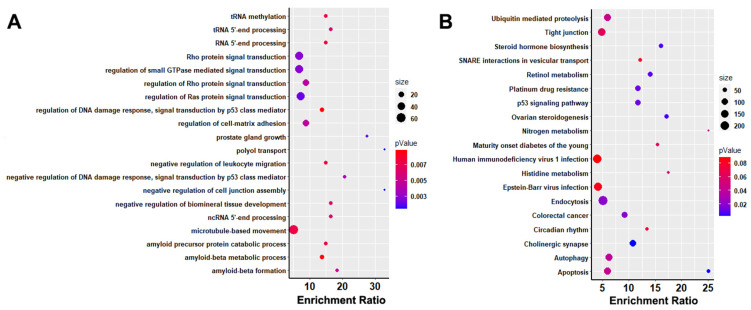
GO and KEGG enrichment analysis of selection genes in commercial pig population. (**A**) The bubble plot revealed the top 20 GO biological progress terms of the enrichment analysis. (**B**) The bubble plot indicated the top 20 KEGG pathways of the enrichment analysis.

**Figure 6 animals-13-03727-f006:**
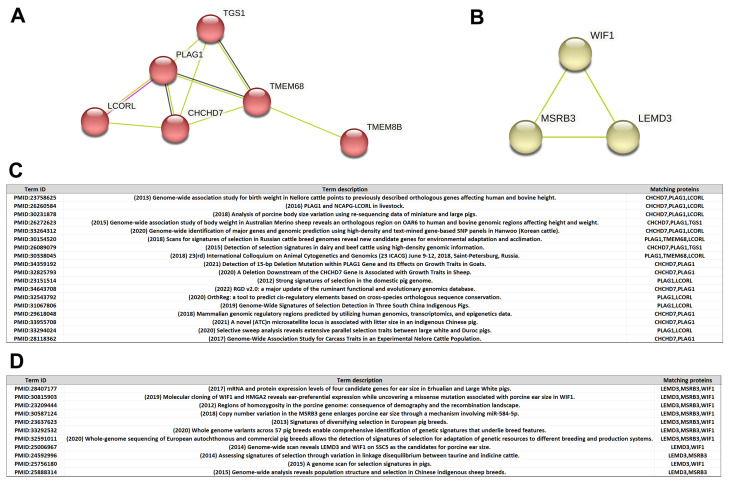
Two PPI subnetworks of selected genes in commercial and indigenous Shanghai pig breeds. (**A**) The PPI subnetwork (consisting of six proteins) subjected to selection in commercial pigs. (**B**) The PPI subnetwork (composed of three proteins) selected in the indigenous Shanghai pig population. (**C**) Genes in the PPI subnetwork obtained in commercial swine populations are strongly correlated with the body size of the livestock and appear in several studies of selection signaling, with the names of the relevant studies in the table. (**D**) Three genes in the PPI subnetwork obtained from an indigenous pig population in Shanghai are closely related to ear traits in livestock, and the names of the related studies are given in the table.

**Figure 7 animals-13-03727-f007:**
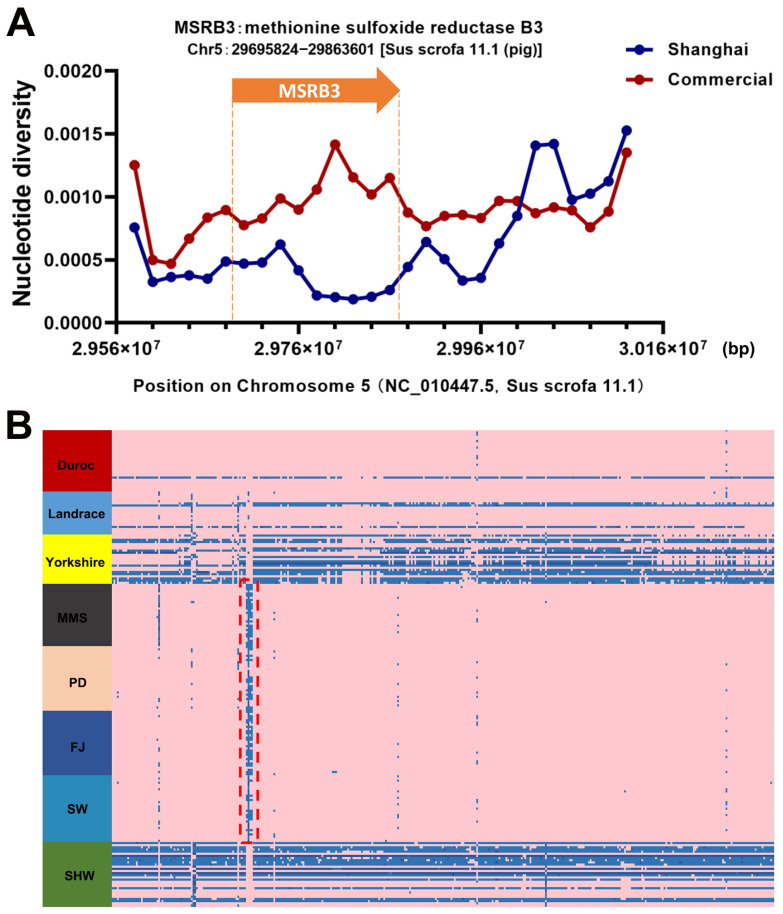
Gene polymorphisms of *MSRB3* of eight pig breed individuals. (**A**) The nucleic acid polymorphism (π) of *MSRB3* in the indigenous Shanghai pig breed population compared to the commercial pig population. (**B**) The haplotype heatmap of *MSRB3* was constructed using these 386 SNPs as columns and the sample individuals as rows, with the genotypes of SNPs consistent with the reference genome indicated in pink (0/0), and the blue parts indicating the loci with polymorphic genotypes (0/1 and 1/1).

**Table 1 animals-13-03727-t001:** Statistical information on whole-genome sequencing of eight pig breeds.

Breed (Individuals)	Average_Clean_Reads	Average_Align_Rate	Average_Depth (×)	Coverage (%)	Coverage_5× (%)
Duroc (31)	335,518,499	99.51	12.00	97.51	87.68
Landrace (21)	382,512,941	99.58	15.25	97.09	88.65
Yorkshire (24)	347,694,640	99.49	12.29	97.32	87.33
FJ (30)	214,356,978	99.80	12.33	97.27	90.78
SW (31)	227,504,591	99.79	13.11	97.40	91.72
MMS (29)	234,489,784	99.43	13.35	97.28	90.65
PD (30)	217,872,738	98.65	12.30	97.18	90.24
SHW (30)	224,392,603	99.66	12.85	97.45	90.87

## Data Availability

The WGS data are publicly available and are deposited here: National Genomics Data Center (https://ngdc.cncb.ac.cn/gsa (accessed on 17 October 2023)) of China, with accession numbers of GSA: CRA012827.

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
