# Peer review of "Genomic Dissection through Whole-Genome Resequencing of Five Local Pig Breeds from Shanghai, China"

_animals, 2023, doi:10.3390/ani13233727_

Round 1

Reviewer 1 Report

Comments and Suggestions for Authors

Its a comprehensive study, comparing the genomes of local and commercial breeds. 

Document is well written, methods clearly explained, quality results provided and nicely discussed. 

However, I have noticed few issues, please consider. 

1. Line 89-91. please provide the criteria, how the relationship (29-31 unrelated individuals from each breed) of selected individuals was confirmed? 

2. Results: In this section, unnecessary explanation of methods is provided. 

3. Discussion: This section is more like a result presentation, very limited reasoning/ justifications of findings. 

4. Concise the conclusions, seems more like an abstract. 

Author Response

Thank you for your valuable comments. I have revised the manuscript one by one on the issues and suggestions, please see the attachment.

Reviewer 2 Report

Comments and Suggestions for Authors

Research on the genetic structure of native pig breeds in the context of increasing homozygosity in commercially maintained pigs is highly important. Typically, native pig breeds have numerous advantages, such as disease resistance and adaptability to challenging environmental conditions, as well as superior quality of the obtained product. Therefore, special breeding programs should be implemented for these pig breeds. It would be advisable to provide in the introduction or discussion section a few examples of other native pig breeds maintained in different countries.

I'm paying attention to a few minor comments – below:

Linia 373 “…hand, The regulatory network formed by the seven genes (TGS1, PLAG1, CHCHD7…”

Linia 389 “…larger the ear size. age in large white pigs, and that individuals with higher expression…”

I note the need for standardizing the reference:

1. Cesar, A., A. Silveira, P. Freitas, E. Guimaraes, D. Batista, L. Torido, F. V. Meirelles and R. Antunes. "Influence of chinese breeds on pork quality of commercial pig lines." Genet. Mol. Res 9 (2010): 727-33.

2. Zhou, R., S. T. Li, W. Y. Yao, C. D. Xie, Z. Chen, Z. J. Zeng, D. Wang, K. Xu, Z. J. Shen and Y. Mu. "The 452 meishan pig genome reveals structural variationmediated gene expression and phenotypic divergence underlying asian pig domestication." Molecular Ecology Resources 21 (2021): 2077-92.

35. Reimer, C., C.-J. Rubin, A. R. Sharifi, N.-T. Ha, S. Weigend, K.-H. Waldmann, O. Distl, S. D. Pant, M. Fredholm and M. Schlather. "Analysis of porcine body size variation using re-sequencing data of miniature and large pigs." Bmc Genomics 19 (2018): 1-17.

36. Utsunomiya, Y. T., A. S. Do Carmo, R. Carvalheiro, H. H. Neves, M. C. Matos, L. B. Zavarez, A. M. Pérez O’Brien, J. Sölkner, J. C. McEwan and J. B. Cole. "Genome-wide association study for birth weight in Nellore cattle points to previously described orthologous genes affecting human and bovine height." BMC genetics 14 (2013): 1-12.

Author Response

Thank you for your valuable comments, I have revised the manuscript one by one on the issues and suggestions, please see the attachment

Reviewer 3 Report

Comments and Suggestions for Authors

In this article, the authors analyzed five Shanghai local pig breeds by whole genome resequencing and found obvious genetic differences between Shanghai local pig breeds and three commercial pig breeds, namely, Duroc, Long White and Yorkshire. And among the analyzed genes and polymorphisms, seven genes closely related to animal growth traits were found, which provided a molecular genetic basis for the conservation and utilization of Shanghai local pig breed resources. However, the manuscript's readability could be much improved to better convey the importance of this study. With editing and some minor revisions, this manuscript will be suitable for publication

In the Abstract part: It is recommended that the full or abbreviated names of breeds of pigs be standardized.

1.     In the article, it is best to have 4-6 keywords, and the existing keywords in the manuscript are not concise enough.

2.     It is recommended that the titles following the figure notes and table headings not be bolded.

3.     In the figure note, replace "A." with "(A)" and bold it.

4.     Line 141 and Line 240, the subscripts in "(θπ)", "Log2θπ" and "FST" are harmonized throughout.

5.     Line 256, additional relevant material or experiments were suggested to demonstrate that CYP19A1 and CYP19A3 are enriched in the pathway.

6.     In this article, the grammar should be revised.

Comments on the Quality of English Language

Moderate editing of English language required

Author Response

(The authors gave the same response as above.)
